# Hollow Nanospheres Organized by Ultra-Small CuFe₂O₄/C Subunits with Efficient Photo-Fenton-like Performance for Antibiotic Degradation and Cr(VI) Reduction

**Dazhi Sun [1], Jiayi Yang [1], Feng Chen [2], Zhe Chen [1,\*] and Kangle Lv [3,\*]**

[1]  School of Material Science and Technology, Jilin Institute of Chemical Technology, Jilin City 132022, China; sundazhi@jlict.edu.cn (D.S.); a4674360@sina.com (J.Y.)

[2]  Jilin Petrochemical Company Organic Synthetic Plants, Jilin City 132021, China; chenfengcf666@163.com

[3]  College of Resources and Environmental Science, South-Central Minzu University, Wuhan 430074, China

\*  Correspondence: chenzhe0809@foxmail.com (Z.C.); lvkangle@mail.scuec.edu.cn (K.L.)

**Abstract:** Hollow transition metal oxides have important applications in the degradation of organic pollutants by a photo-Fenton-like process. Herein, uniform, highly dispersible hollow CuFe₂O₄/C nanospheres (denoted as CFO/C-PNSs) were prepared by a one-pot approach. Scanning electron microscope (SEM) and transmission electron microscope (TEM) images verified that the CFO/C-PNS catalyst mainly presents hollow nanosphere morphology with a diameter of 250 ± 30 nm. Surprisingly, the photodegradation test results revealed that CFO/C-PNSs had an excellent photocatalytic performance in the elimination of various organic contaminants under visible light through the efficient Fenton catalytic process. Due to the unique hollow structure formed by the assembly of ultra-small CFO/C subunits, the catalyst exposes more reaction sites, improving its photocatalytic activity. More importantly, the resulting magnetically separable CFO/C-PNSs exhibited excellent stability. Finally, the possible photocatalytic reaction mechanism of the CFO/C-PNSs was proposed, which enables us to have a clearer understanding of the photo-Fenton mechanism. Through a series of characterization and analysis of degradation behavior of CFO/C-PNS samples over antibiotic degradation and Cr(VI) reduction, •OH radicals generated from H₂O₂ decomposition played an essential role in enhancing the reaction efficiency. The present work offered a convenient method to fabricate hollow transition metal oxides, which provided impetus for further development in environmental and energy applications. **Highlights:** Novel hollow CuFe₂O₄/C nanospheres were prepared by a facile and cost-effective method. CuFe₂O₄/C exhibited excellent photo-Fenton-like performance for antibiotic degradation. Outstanding photocatalytic performance was attributed to the specific hollow cavity-porous structure. A possible mechanism for H₂O₂ activation over hollow CuFe₂O₄/C nanospheres was detailed and discussed.

**Keywords:** hollow CuFe₂O₄/C nanospheres; photo-Fenton; photodegradation of various pollutants; photocatalytic reaction mechanism

## 1. Introduction

In recent years, the discharge of various organic pollutants has caused irreversible repercussions to natural ecosystems and human health [1–4]. Many methods are used for solving these environmental problems, such as photocatalysis, physical adsorption and advanced oxidation [5–17]. Among them, photocatalytic Fenton oxidation [18–20] has attracted more and more attention due to its advantages of simple operation, high efficiency, no secondary pollution and effective removal of environmental pollutants [21–25]. Nevertheless, the traditional homogenous Fenton processes include obvious and inevitable shortcomings, such as production of iron containing sludge, low pH (pH < 3.0), large amount of H₂O₂, requirement for secondary treatment and difficult regeneration of the catalyst, which have greatly restricted their wide applications. To solve this problem, a

"Fenton-like" reaction, which can surmount the above-mentioned adverse conditions as, for example, the reduction of $H_2O_2$ by modifying the process going towards the photo-Fenton reaction, has been considered as one of the most effective methods [26–34]. Therefore, the key point is to exploit suitable semiconducting, heterogeneous and Fenton-like catalysts with a wide pH range, high reusability, excellent activity and good stability. Magnetic spinel ferrite nanoparticles, including $MFe_2O_4$ (M = Mn, Fe, Co, Ni, Cu, Zn) as one of the promising materials, have received much attention due to their moderate costs, recyclability, and magnetic and electrical properties [35–38]. Among them, a p-type semiconductor copper ferrite ($CuFe_2O_4$), an important metal oxide (band gap of 1.9 eV), has been applied in various fields owing to its superior magnetic, optical and catalytic properties [39–42]. In addition, there are numerous approaches that are effective methods to prepare copper ferrite, such as ball-milling, microwave and solvothermal methods [43–49]. Recently, most of the $CuFe_2O_4$ reports have been widely studied in the photocatalysis field and have considered $CuFe_2O_4$ as a superior candidate for wastewater due to its advantages, such as easy synthesis, inexpensive cost for production, good optical property, excellent catalytic activity and simple magnetic recoverability [50–56]. Therefore, in order to obtain photocatalysts with efficient performance, it is of great importance to develop convenient strategies for synthesizing various morphologies of $CuFe_2O_4$ and $CuFe_2O_4$-based photocatalysts, such as hollow mesoporous $CuFe_2O_4$, core-shell $CuFe_2O_4@C_3N_4$, $CuFe_2O_4@SiO_2$ nanofibers and so on [57–63]. The hollow spherical structure of photocatalysis has a large specific surface area; therefore, it can expose more active sites to improve the photocatalytic agent. However, thus far, exploiting a simple strategy to increase photocatalyst activity and meet the requirements of practical applications with a unique nanostructure design is still a challenge.

In this work, we report a one-pot synthetic calcination method for the first time to fabricate uniform, highly dispersible hollow CFO/C-PNSs assembled by ultra-small CFO/C subunits through a simple and cost-effective strategy. The hollow microspheres have a high specific surface area and excellent mass transfer performance, which is beneficial to improve the photocatalytic performance. The high specific surface area provides more active sites and allows more pollutant molecules to adsorb. Meanwhile, hollow microspheres enhance the utilization efficiency of incident visible light by enhancing the reflection and scattering of light. The as-synthesized CFO/C-PNSs exhibited a markedly photocatalytic performance in photo-Fenton-like reactions for the heterogeneous activation of $H_2O_2$ with irradiation of visible light, which can degrade and mineralize oxytetracycline (OTC), norfloxacin (NFX), tetracycline (TCH), rhodamine B (RhB), methyl orange (MO) and Cr(VI) reduction in solution.

## 2. Experimental Procedures

Specific experiment and characterization methods can be found in supporting information. The schematic illustration for the controllable synthetic of the CFO/C-PNSs with a relatively simple process is illustrated in Figure 1. Firstly, PAA-NH$_4$ with a globose structure was prepared by the addition of $NH_3 \cdot H_2O$ to an isopropyl alcohol aqueous solution. After that, a certain amount of $FeCl_2 \cdot 4H_2O$ was added into the PAA-NH$_4$ aqueous solution and $Fe^{2+}$ was further anchored to the surface of an anionic polymer (PAA) at random to form $Fe(OH)_3$/PAA solution in the alkaline conditions due to the charge of the interfacial energy of the synthetic system. Then, $CuCl_2 \bullet 2H_2O$ was added, and $Cu^{2+}$ was readily hydrolyzed into $Cu(OH)_2$ nanoparticles in a weak base environment and further aggregated on the outer and inner surfaces of the $Fe(OH)_3$/PAA layer to synthesize $Fe(OH)_3$/PAA/$Cu(OH)_2$ nanoparticles. More importantly, due to the electrostatic repulsion of the carboxyl groups, the $Fe(OH)_3$/PAA/$Cu(OH)_2$ in solution had high dispersibility. Lastly, the $CuFe_2O_4$/C-PNSs with hollow nanostructures were fabricated by a heating treatment at 400 °C for 120 min under $N_2$ gas protection, and named as CFO/C-PNSs.

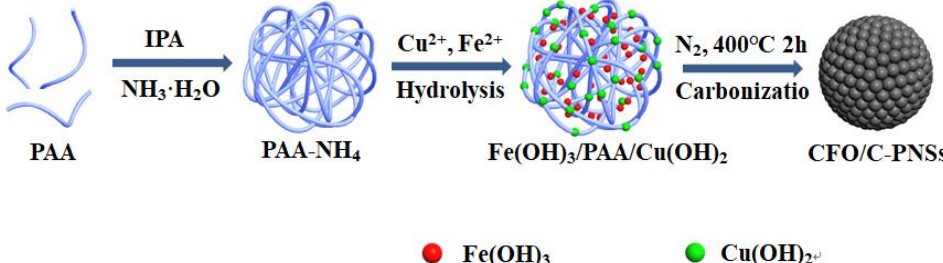

**Figure 1.** Schematic illustrating the synthesis process of the CFO/C-PNSs sample.

## 3. Results and Discussion

### 3.1. Structure and Composition Characterization

The phase structures and specific composition of synthesized samples were confirmed by X-ray diffraction (XRD), as presented in Figure 2. The main diffraction peaks of CFO/C-PNSs and CFO-PNSs are fully consistent with the spinel $CuFe_2O_4$ standard file (PDF#77-0010) [64], with diffraction peaks of the synthesized samples at 30.18°, 35.54°, 43.20°, 53.60°, 57.14°, 62.74° and 74.24° corresponding to (220), (311), (400), (422), (511), (440) and (533). The sharpened and intense diffraction peaks manifest the highly crystalline nature of sample. Obviously, no peaks of impurities can be found, implying the $Fe(OH)_3/PAA/Cu(OH)_2$ is utterly transformed to phase-pure $CuFe_2O_4/C$ after calcination at high temperature under $N_2$ gas protection. Accordingly, the consequences of XRD patterns illustrate that the $CuFe_2O_4/C$ was successfully prepared using calcination treatment under an atmosphere of $N_2$.

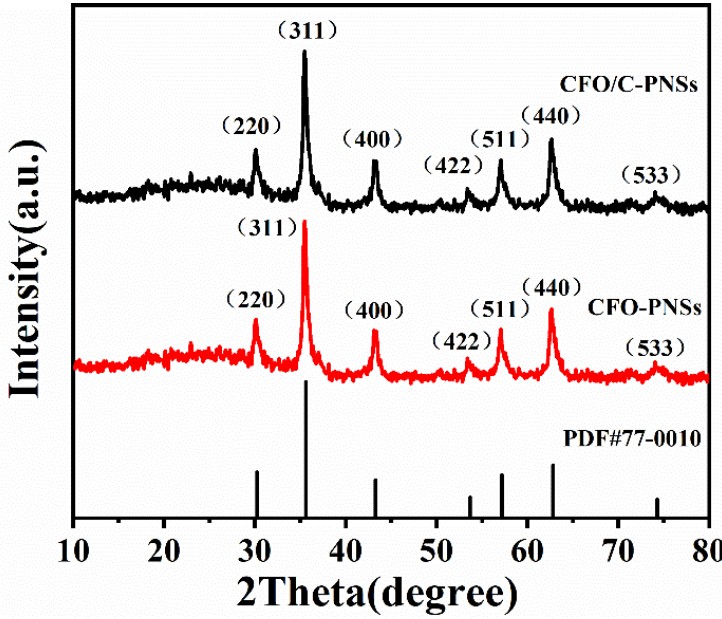

**Figure 2.** XRD spectra of CFO/C-PNS and CFO-PNS samples.

The information of morphology and nanostructure of the as-synthesized CFO/C-PNSs were provided by using a scanning electron microscope (SEM), transmission electron microscope (TEM) and high-resolution TEM (HRTEM) (Figure 3). Figure 3a demonstrates how the SEM image exhibits highly and uniformly diffused rough spherical structures with a diameter size range of 250 ± 30 nm, which are assembled with an ultrafine $CuFe_2O_4/C$ subunit with a diameter of about 5 nm. Accordingly, these images imply a unique pomegranate-like nanostructure which can provide more active sites during the photocatalytic process. More details about the morphology and nanostructure of CFO/C-PNSs were examined by the TEM image. Figure 3b shows that the CFO/C-PNS nanoparticles have a perfectly spherical

shape corresponding to the SEM image. Moreover, the HRTEM image of CFO/C-PNSs in Figure 3c presents the single pomegranate-like CFO/C-PNSs consisting of a good amount of ultra-small CFO/C-PNS subunits. Additionally, the lattice fringes of 0.36 nm are wider than the (311) planes of CuFe2O4 (JCPDS 77-0010); this may be due to the lattice change caused by the incorporation of carbon in $CuFe_2O_4$ (the inset of Figures 3c and S1). Finally, the elemental mapping images of single CFO/C-PNSs in Figure 3e–h show the Fe (carmine), Cu (yellow), O (red) and C (blue) elements were uniformly distributed on the pomegranate-like CFO/C-PNS nanosphere.

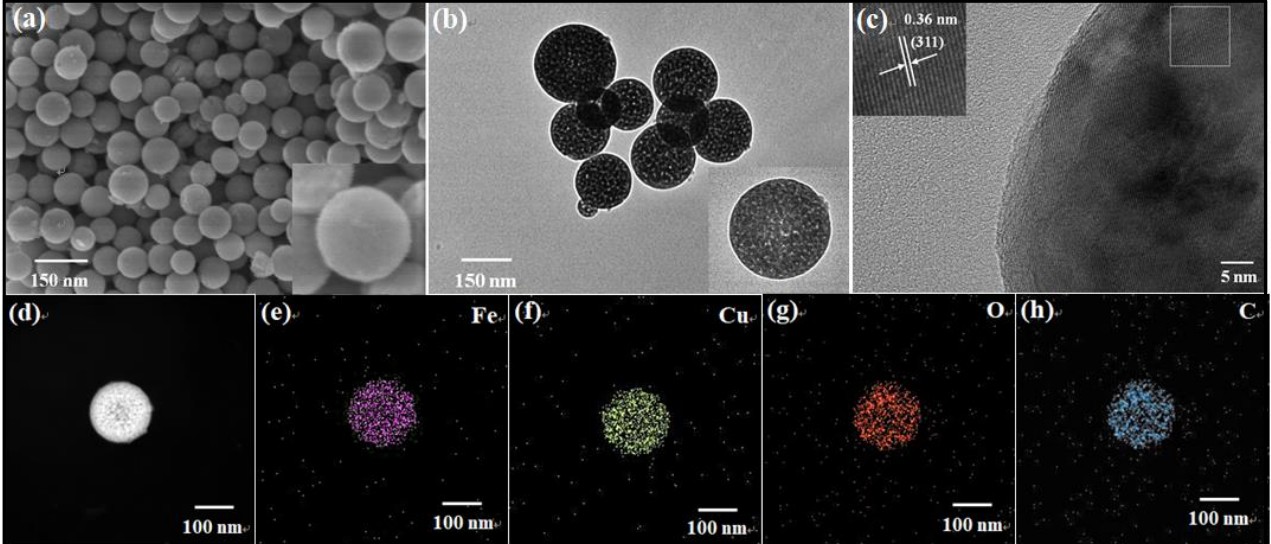

**Figure 3.** The SEM (**a**), TEM (**b**) and HRTEM (**c**) images for CFO/C-PNS sample; (**d**) elemental mappings of Fe (**e**), Cu (**f**), O (**g**) and C (**h**) elements of CFO/C-PNS sample.

The nitrogen sorption analysis was used to measure porosity of CFO/C-PNSs and CFO-PNSs. As presented in Figure 4a,b, the specific surface areas of photocatalyst samples were calculated with the multipoint Brunauer–Emmett–Teller (BET) method; the specific surface areas of CFO/C-PNSs and CFO-PNSs are 179.7 m$^2$/g and 71.8 m$^2$/g, respectively, and the total pore volumes of these samples are about 0.21 cm$^3$/g and 0.14 cm$^3$/g, respectively. In addition, the CFO/C-PNSs display the IV-type isotherms with the H3-type hysteresis loops, implying that they possess the appropriate micro-mesoporous feature. Such results suggest that CFO/C-PNSs could expose more active sites and many efficient transport pathways, which is beneficial for the photocatalytic performance. Figure 4c presents the optical properties of prepared samples which were measured by the UV–vis diffuse reflectance spectra. It is clearly displayed that the CFO/C-PNSs show stronger absorption intensity than CFO-PNSs from UV to visible light, indicating that compared to CFO-PNSs, CFO/C-PNSs can improve the utilization efficiency of solar energy, and enhance the photocatalytic activity of the sample under illumination. Additionally, the band gaps of CFO/C-PNSs and CFO-PNSs can be evaluated via the Kubelka–Munk function, as illustrated in Figure 4d [65]. Notably, the estimated band gaps of CFO/C-PNSs and CFO-PNSs are calculated to be about 1.6 eV and 1.9 eV, respectively, which indicate that CFO/C-PNSs enable more efficient usage of solar energy than the pure CFO-PNSs.

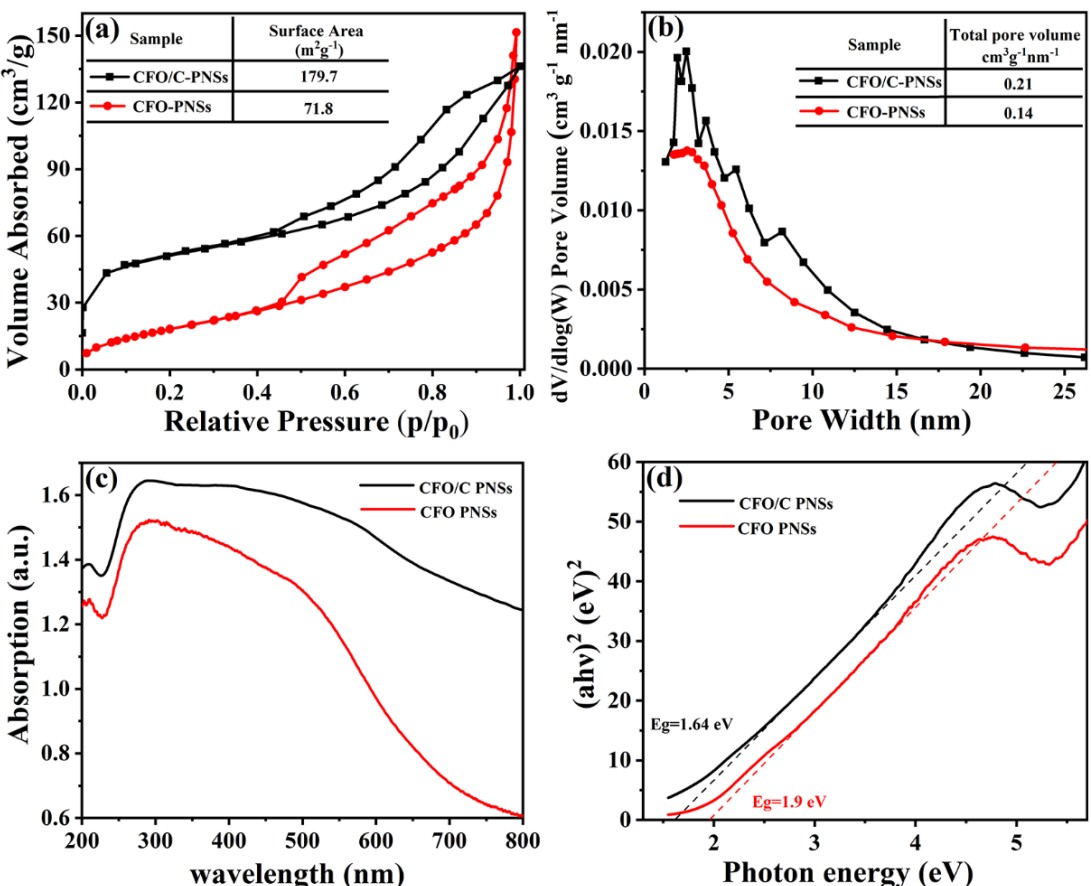

**Figure 4.** (**a**) $N_2$ adsorption–desorption isotherms and (**b**) pore size distribution, (**c**) UV–vis DRS spectra and (**d**) the band gap energies (Eg) of CFO/C-PNSs and CFO-PNSs.

### 3.2. Evaluation of Photocatalytic Performance

　　The photo-Fenton-like perfomance of CFO/C-PNSs and CFO-PNSs was evaluated by the photocatalytic degradation of MO, OTC, CR, RhB, TCH, and NFX, and the reduction of $K_2Cr_2O_7$ in the presence of $H_2O_2$. (Figures 5 and S2). Before the photo-Fenton-like reaction, a dark adsorption test was carried out for 30 min to reach an equilibrium adsorption state between the catalysts and contaminants. Figures 5a,e and S2 show that almost all various antibiotics were degraded by the CFO/C-PNSs after visible-light irradiation for 1 h in the presence of $H_2O_2$ (200 µL); under the same conditions, the degradation performance of CFO-PNSs was obviously inferior. More importantly, the CFO-PNSs exhibited the highest photo-Fenton degradation efficiencies of TCH (98.75%), NFX (97.13%), OTC (92.36%), MO (90.60%), RhB (99.1%) and photoreduction Cr(VI) (90.42%) within 60 min, which suggested that it has great potential in water treatment in the future.

　　For a better catalytic efficiency comparison of the CFO/C-PNSs and CFO-PNSs, kinetic analyses of the degradation of various antibiotics are described in Figure 5c,d through a pseudo-first-order reaction model and the reaction kinetic rate was 0.05183 $min^{-1}$ for TCH, 0.04607 $min^{-1}$ for NFX, 0.0409 $min^{-1}$ for OTC, 0.03138 $min^{-1}$ for MO, 0.08155 $min^{-1}$ for RhB, and 0.08155 $min^{-1}$ for Cr(VI), but the reaction kinetic rates of CFO-PNSs for all various antibiotics were lower. The above results indicate that the hollow CFO/C-PNSs have superior photo-Fenton-like catalytic activity for the MO, OTC, RhB, TCH, NFX and photoreduction of Cr(VI), which is attributed to the ultrafine $CuFe_2O_4/C$ subunits and the larger specific surface area of CFO/C-PNSs, namely, each ultrafine $CuFe_2O_4/C$ subunit can effectively and directly work on the photocatalytic reaction, which can expose more active sites and absorb more pollutants on its surface, and further dramatically shorten reaction pathways for photo-Fenton-like degradation experiments. Moreover, considering

the practical application, the magnetic property of CFO/C-PNSs was revealed by applying a vibrating sample magnetometer at room temperature in an applied magnetic field up to 10,000 Oe (Figure 5f). The CFO/C-PNS catalyst showed an excellent ferromagnetic property with a saturation magnetization (Ms) of 99.4 emu/g. Moreover, the complete separation of CFO/C-PNSs from the solution can be achieved with an external magnet within 15 min, which is quite qualified for the magnetic separation and recycling of the photo-Fenton catalyst and further ascertaining its application in the photodegradation of organic pollutants.

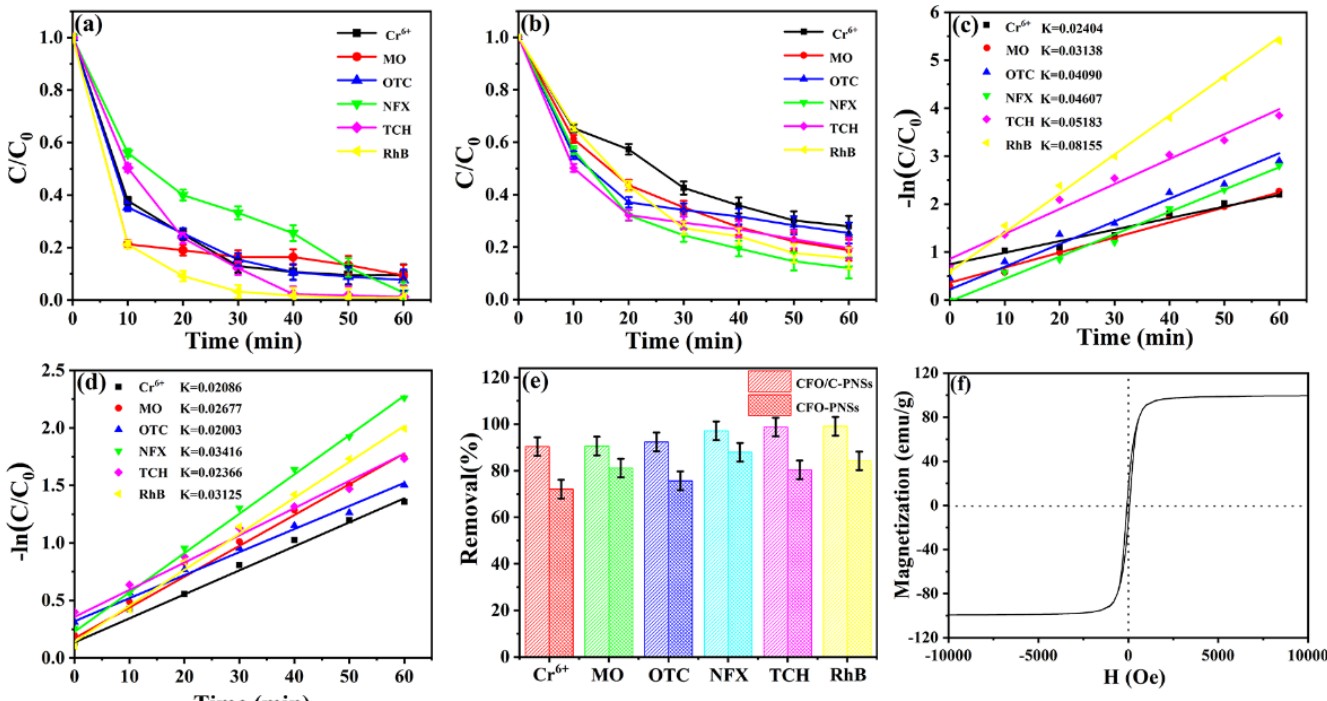

**Figure 5.** The visible-light photocatalytic performance of (**a**) CFO/C-PNSs and (**b**) CFO-PNSs for photoreduction of Cr(VI) and degradation of MO, OTC, NFX, TCH and RhB. The corresponding reaction rate constant k is shown as (**c**,**d**), respectively. (**e**) The removal rate of CFO/C-PNSs and CFO-PNSs for Cr(VI) MO, OTC, NFX, TCH and RhB. (**f**) Hysteresis loop diagram of CFO/C-PNS photocatalysts.

The surface chemical and valence states of CFO/C-PNSs and CFO-PNSs were characterized by X-ray photoelectron spectroscopy (XPS) (Figure 6). From high-resolution XPS analysis of Cu 2p, as depicted in Figure 6a, the Cu 2p peaks were fitted into four peaks at 934.8 and 954.73 eV, corresponding to binding energies of the Cu $2p_{3/2}$ and Cu $2p_{1/2}$ doublet, as well as three of the shakeup satellites at 942.49, 943.2 and 962.82 eV [66,67]. The two-peak separation value ($\Delta E$) was found to be about 20 eV, ascribing to the presence of the $Cu^{2+}$ oxidation state in CFO/C-PNS peaks. In the Fe 2p XPS spectrum of CFO/C-PNSs (Figure 6b), two peaks appearing at 711.74 and 725.36 eV were assigned to the characteristic binding energy of Fe 2p $_{3/2}$ and Fe 2p $_{1/2}$, and the two small satellite peaks were ascribed to the oxidation state of $Fe^{3+}$ in the CFO/C-PNSs [68–70]. For the O 1s spectrum, the three types of O species were detected at binding energies of 532.20, 529.93 and 530.59 eV in the CFO/C-PNSs (Figure 6c). The peaks at the lower binding energies of 529.93 eV and 532.20 eV are ascribed to the lattice oxygen $O^{2-}$ from Cu–O and Fe–O linkages [71]. The intermediate binding energy peak was 530.59 eV, which is related to $O^{2-}$ in the oxygen-deficient regions, confirming the occurrence of oxygen vacancies in the CFO/C-PNSs [72]. Furthermore, the high binding energy peak at 532.20 eV is assigned to absorbed oxygen species [73]. The C 1s spectrum (Figure 6d) of CFO/C-PNSs exhibits two

stronger peaks, at 284.33 eV and 284.95 eV, which are assigned to C-C/C=C and C-O/C=O, respectively [74]. Hence, the XPS results further confirmed that the normal valence state of Cu and Fe elements in the CFO/C-PNSs are +2 and +3, respectively. In addition, the results further confirmed that spinel structure CFO/C-PNSs have been successfully synthesized the same as with the XRD results.

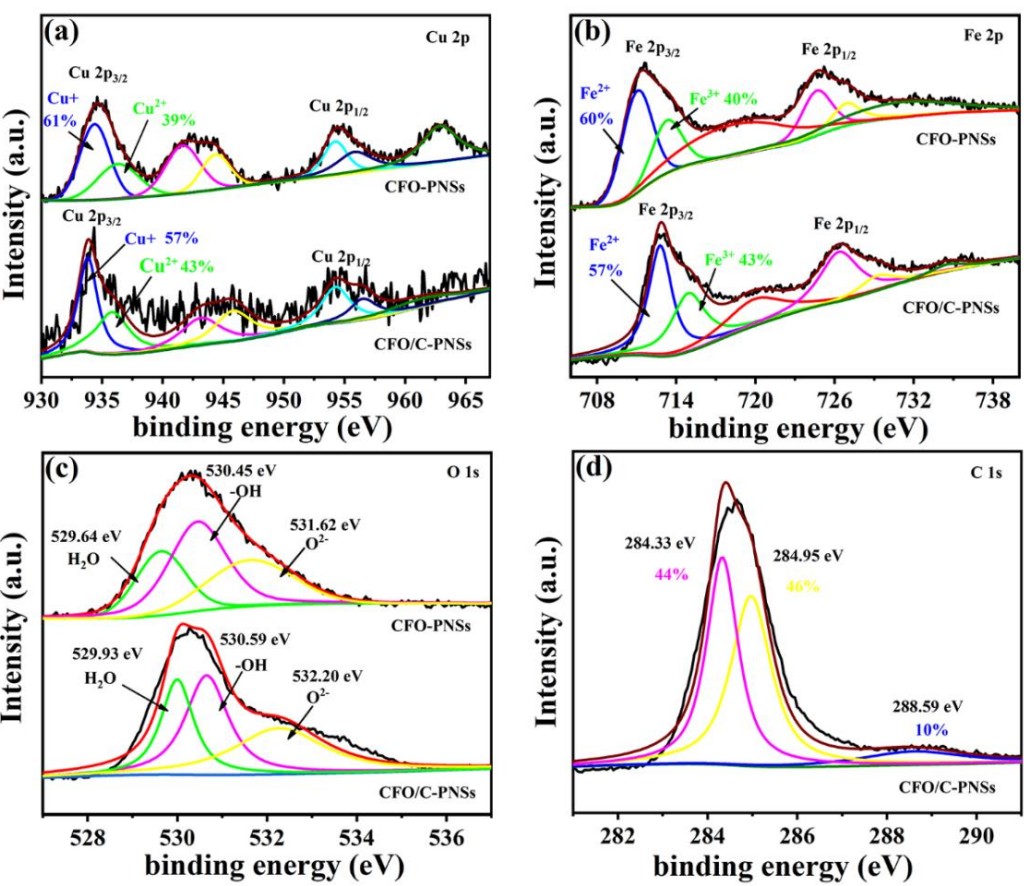

**Figure 6.** XPS spectra of CFO/C-PNS and CFO-PNS samples: (**a**) Cu 2p, (**b**) Fe 2p, (**c**) O 1s, (**d**) C 1s.

To reveal the role the CFO/C-PNS catalyst played in the photo-Fenton degradation reaction, we carried out a series of comparative tests as shown in Figures 7 and S3. Notably, the effect of $H_2O_2$ alone on photoreduction degradation of various pollutants (MO, OTC, NFX, TCH, RhB and photoreduction of $K_2Cr_2O_7$ degradation) was small, suggesting that $H_2O_2$ alone was ineffective. Under visible light, contaminants of the system can be removed only within 1 h in the presence of $H_2O_2$, suggesting that both $H_2O_2$ and visible light can form $\bullet O_2^-$ radicals and generate more and more $\bullet OH$, further significantly enhancing the rate of the photo-Fenton degradation. Moreover, the CFO/C-PNSs +Vis + $H_2O_2$ show much higher removal rates (98.75% of TCH, 97.13% of NFX, 92.36% of OTC, 90.60% of MO, 99.1% of RhB and 90.42% of photoreduction of $K_2Cr_2O_7$ degradation) than CFO/C-PNSs + Vis (39.99% of MO, 85.60% of OTC, 61.97% of RhB, 32.62% of TCH, 20.74% of NFX and 10.77% of photoreduction of $K_2Cr_2O_7$ degradation), due to the photogenerated electrons that can expedite the generation of $\bullet OH$ and promote the regeneration of $Fe^{2+}$. Similarly, CFO/C-PNSs + $H_2O_2$ show a much lower performance than the CFO/C-PNSs + Vis, because the presence of visible light is required for efficient photo-Fenton degradation of various contaminants (Figure S3). The aforementioned experiment results indicate that both $H_2O_2$ and visible light are indispensable conditions to obtain satisfactory results from photo-Fenton activity.

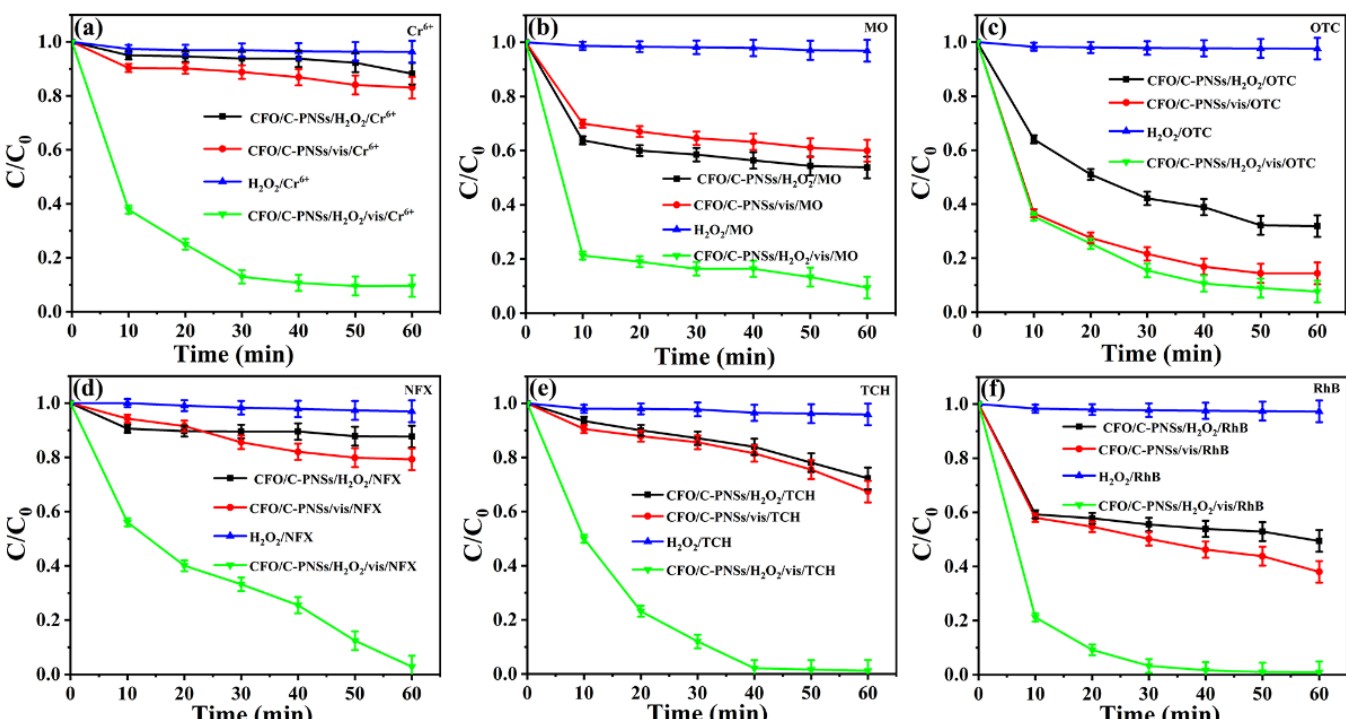

**Figure 7.** The photocatalytic degradation of (**a**) Cr(VI), (**b**) MO, (**c**) OTC, (**d**) NFX, (**e**) TCH and (**f**) RhB by CFO/C-PNSs with different conditions under visible-light irradiation.

For evaluating the stability and the practical application of the obtained CFO/C-PNSs, the recycling photo-Fenton degradation experiments toward a contaminant were implemented with irradiation of visible light in the presence of $H_2O_2$. As plotted in Figure 8a,b, the performance of CFO/C-PNSs did not display significant deterioration over three successive runs, suggesting its outstanding stability in aqueous solution. The slight reduction in degradation rate is due to some inevitable factors in the test, such as the passivation of the catalyst surface or the loss of the catalyst in the recycling and rinsing process. These results suggest that the CFO/C-PNSs have good stability and durability in a photocatalysis-Fenton reaction, which would be a potential way to treat wastewater.

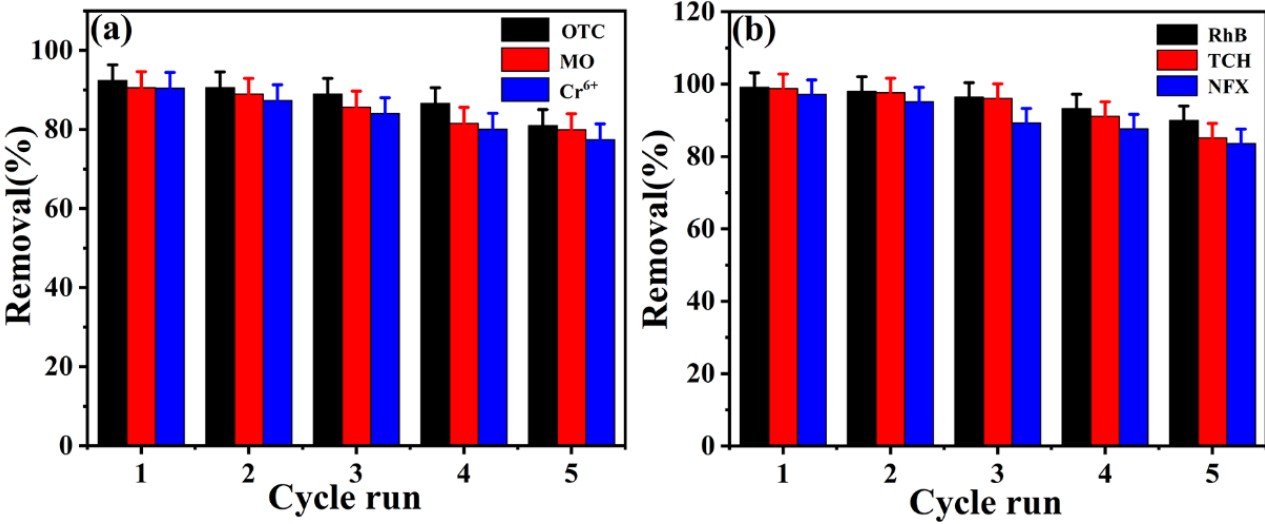

**Figure 8.** Cyclic experimental degradation diagram of CFO/C-PNS photodegradation of (**a**) OTC, MO and Cr(VI), and (**b**) RhB, TCH and NFX.

To investigate different factors of the photo-Fenton degradation activity of CFO/C-PNSs, taking tetracycline hydrochloride as an example, a series of comparative experiments including different volumes of $H_2O_2$, pH value and catalyst concentration were performed (Figure 9). $H_2O_2$ is a significant parameter in the photocatalytic process. Figure 9a illustrates the photocatalytic degradation curves of THC for the different $H_2O_2$ concentrations, while holding other conditions constant. In the contrast experiment, the CFO/C-PNS catalyst exhibited a very high Fenton-like activity with increasing $H_2O_2$ volume from 0 to 200 µL and the TCH degradation efficiency considerably enhanced, ultimately obtaining a maximum efficiency of 98.75%. Nevertheless, its activity decreased slightly with the increase in the $H_2O_2$ dose (500 µL), which might be due to an excess of $H_2O_2$ that can clean up the generated •OH radicals and adversely affect the reaction, namely, superabundant $H_2O_2$ could produce hydroperoxyl radicals (HO$_2$•, $H_2O_2$+•OH→$H_2O$+ HO$_2$•) and reduce the availability of •OH owing to the quenching of •OH by competing reactions with $H_2O_2$ and HO$_2$• (•OH + $H_2O_2$ →$H_2O$ + HO$_2$• and •OH + HO$_2$• → $H_2O$ + $O_2$) [75]. Hence, the above experiment results illustrated that an appropriate dose of $H_2O_2$ is conducive to strengthening the photo-Fenton reaction for TCH degradation.

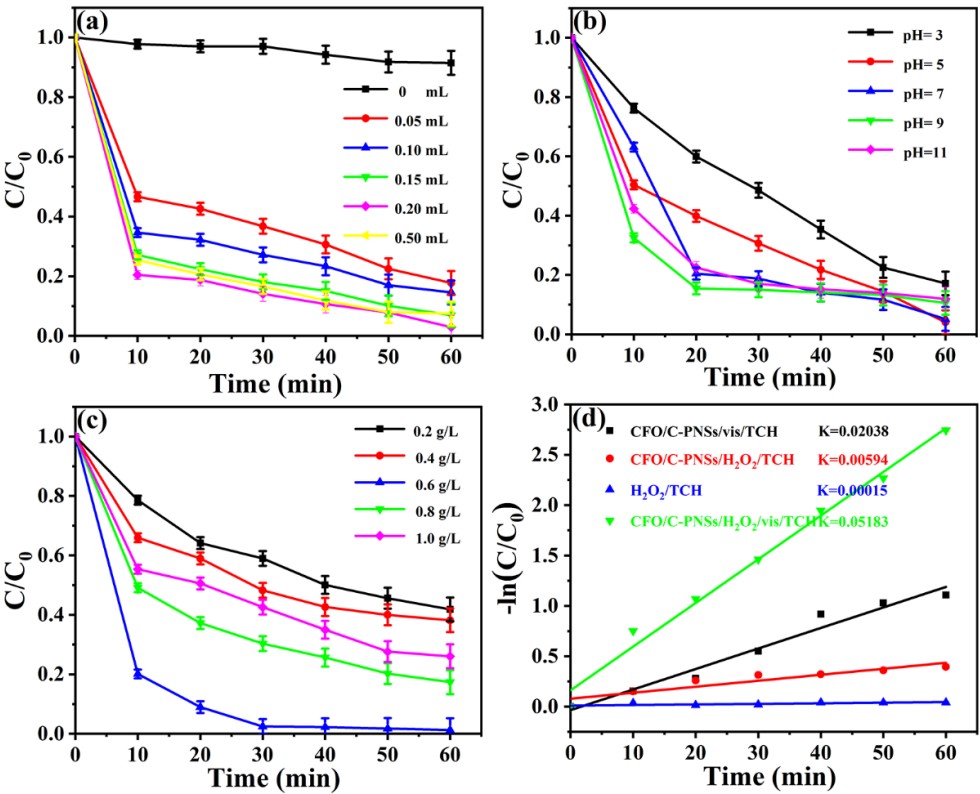

**Figure 9.** (**a**) Photocatalytic degradation of TCH by the CFO/C-PNS samples with different volumes of $H_2O_2$ and (**b**) different values of pH (the amount of $H_2O_2$: 200 µL); (**c**) photocatalytic degradation of TCH with different CFO/C-PNS contents (the amount of $H_2O_2$: 200 µL); (**d**) reaction rate constant k of photocatalytic degradation of TCH with different conditions under visible-light irradiation.

Because the activity of heterogeneous Fenton is very limited at neutral and alkaline pH, we further studied the effect of the initial pH value on degradation of TCH in the photo-Fenton reaction, and the photocatalytic performance of the CFO/C-PNS catalyst was conducted under the same experimental conditions. Figure 9b depicts the degradation efficiency of TCH in the CFO/C-PNS catalyst, which exhibited high activity over a wide pH range (pH 3–11), and the degradation efficiencies were all over 90% due to more •OH produced by the activation of $H_2O_2$. By contrast, the degradation rate slightly decreased with an increasing pH over 9 in this photo-Fenton reaction, which may be due to the generation

of mineral ions during the degradation of TCH that further inhibited the adsorption of ions. The experiment results indicated that the CFO/C-PNSs as a photo-Fenton catalyst were suitable for an extensively wide range of pH, which further illustrated that CFO/C-PNSs are a good candidate for further practical waste treatment. In addition, the photocatalytic degradation curves of THC for different concentrations of the CFO/C-PNS catalyst (i.e., 0.2, 0.4, 0.6, 0.8 and 1 g/L) were further evaluated and kept the other conditions constant, as presented in Figure 9c. Similarly, the degradation efficiency also gradually increased from 58.18 to 61.84% until it reached a maximum of 98.75% within 60 min of irradiation by increasing the concentration of the CFO/C-PNS catalyst. Contrarily, as the amount of photocatalyst increased, the degradation rate decreased gradually when the initial concentration of the catalyst increased to 0.6 g/L. This decrease in the degradation rate is mainly ascribed to the excess photocatalysts, which cause the degree of light scattering and excessive turbidity of the solution, decrease the active sites and retard the light penetration, hence, leading to a reduction in the degradation rates of TCH. The kinetic curves of comparative experiments on degradation of TCH in the photo-Fenton reaction used the $K_{ap}$ value to evaluate the photocatalytic activity. As shown in Figure 9d, the CFO/C-PNS catalyst demonstrates almost 2.54 times the degradation rate (0.05183 $min^{-1}$) with irradiation of visible light in the presence of $H_2O_2$ (CFO/C-PNSs/Vis/$H_2O_2$) as compared to the CFO/C-PNS catalyst under visible-light irradiation (CFO/C-PNSs/Vis), suggesting that $H_2O_2$ is the dominant element for strengthening the photocatalytic property. Meanwhile, the degradation rate of 0.02038 $min^{-1}$ (CFO/C-PNSs/Vis) is larger than that of 0.00594 $min^{-1}$ (CFO/C-PNSs/$H_2O_2$), revealing that both $H_2O_2$ and visible light are responsible for important factors in photo-Fenton degradation processes.

The electrochemical impedance spectroscopy (EIS) test was performed to study the charge transfer rate at the interface with 0.1 M KCl solution containing 5 mM Fe(CN)$_6^{3-}$/$^{4-}$ [75]. As presented in Figure 10a, CFO/C-PNSs have a smaller semicircle radius than CFO-PNSs, and EIS measurements indicated that CFO/C-PNSs exhibit a smaller charge-transfer resistance [76]. CFO/C-PNSs are more prone to electron transfer during RED and have faster interfacial charge transfer and more efficient carrier separation. The Mott–Schottky (M–S) data is shown in Figure 10b, where the band structure of the sample was further analyzed by the Mott–Schottky curve, and the CB of CFO/C-PNSs and CFO-PNSs were about −0.71 eV and −0.83 eV, respectively [77]. The linear sweep voltammetry (LSV) test further verified that the CFO/C-PNS catalyst played a greater role in the photocatalytic process (Figure 10c). Compared with the electrode of CFO-PNSs, the electrode of CFO/C-PNSs had lower overpotential, which was favorable for the RED reaction in the photocatalytic degradation process [78]. In summary, the electrochemical performance of the CFO/C-PNS photocatalyst indicates that it can promote the separation and transmission efficiency of photogenerated carriers.

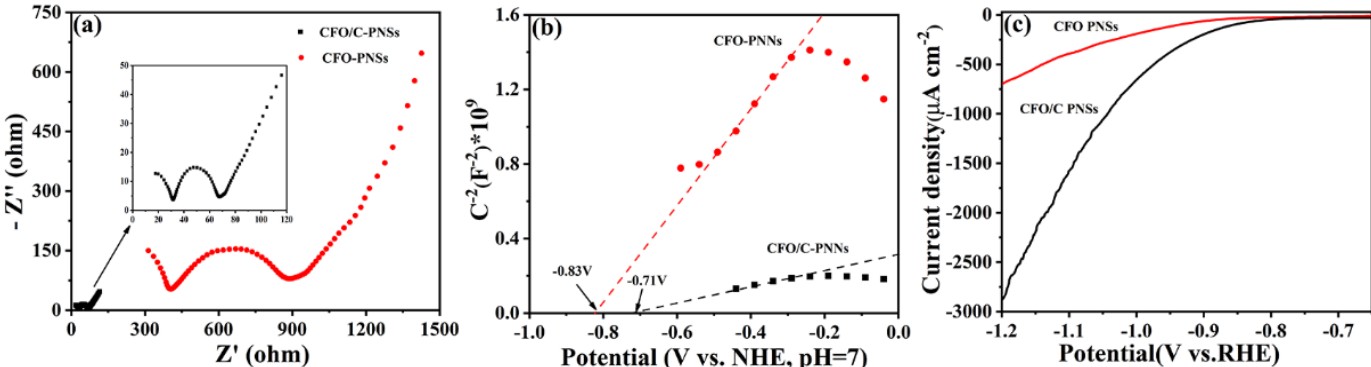

**Figure 10.** (**a**) Electrochemical impedance spectroscopy (EIS); (**b**) Mott–Schottky curves (M–S); (**c**) linear sweep voltammetry curve (LSV).

### 3.3. The Photocatalytic Mechanism

To deeply investigate the mechanism of TCH degradation, the free-radical capture tests and the electron spin response (ESR) spectra were studied to analyze the effects of different scavengers in photo-Fenton degradation experiments. The main oxidative species were detected by isopropanol (IPA), triethanolamine (TEOA) and 4-hydroxy-TEMPO, which were respectively served as hydroxyl radical (•OH), hole ($h^+$) and superoxide radical (•$O_2^-$) quenchers into the photocatalytic reaction systems [79–81] (Figure 11a). It could be observed that the degradation efficiency of TCH was scarcely unchanged upon addition of IPA, which applies CFO/C-PNSs as the representative catalyst, suggesting that •OH is responsible for this photo-Fenton reaction because •OH radicals form via the $H_2O_2$ activation. On the contrary, the degradation rate of TCH was notably decreased by the introduction of 4-hydroxy-TEMPO/TEOA in the solution, revealing that •$O_2^-$ and holes are not the main active intermediates in this photo-Fenton degradation experiment. To further identify this experiment's results, the ESR spin-trap technique was carried out in the photo-Fenton-like reaction process. In the aqueous solution, the •OH and •$O_2^-$ can be easily captured by 5,5-dimethyl-L-pyrroline N-oxide (DMPO) and methanol to form the DMPO-•OH and DMPO-•$O_2^-$ adducts. As depicted in Figure 11b, the CFO/C-PNS sample cannot produce •OH radicals under visible-light irradiation. Notably, the characteristic signals of •OH radicals can be easily detected for the CFO/C-PNS sample under irradiation of visible light in the presence of $H_2O_2$, which convincingly reveals that •OH plays an essential part in the photo-Fenton-like reaction process due to $Fe^{2+}$ and $Fe^{3+}$ reacting with $H_2O_2$ to produce •OH, and the hole reacting with $H_2O_2$ to form •OH radicals [82]. As shown in Figure 11c, the ESR signal intensity of DMPO-•$O_2^-$ adducts for the CFO/C-PNS sample under the condition that adding to $H_2O_2$ is much stronger than that of the CFO/C-PNS sample with a shortage of $H_2O_2$, most probably due to electrons or holes which react with $H_2O_2$ to produce •$O_2^-$ radicals [83]. Apparently, the ESR results further confirm that the hydroxyl radicals (•OH), as main reactive oxidative species, play a major part in the degradation of pollutants, which could effectively boost the photo-Fenton catalytic efficiency of contaminants.

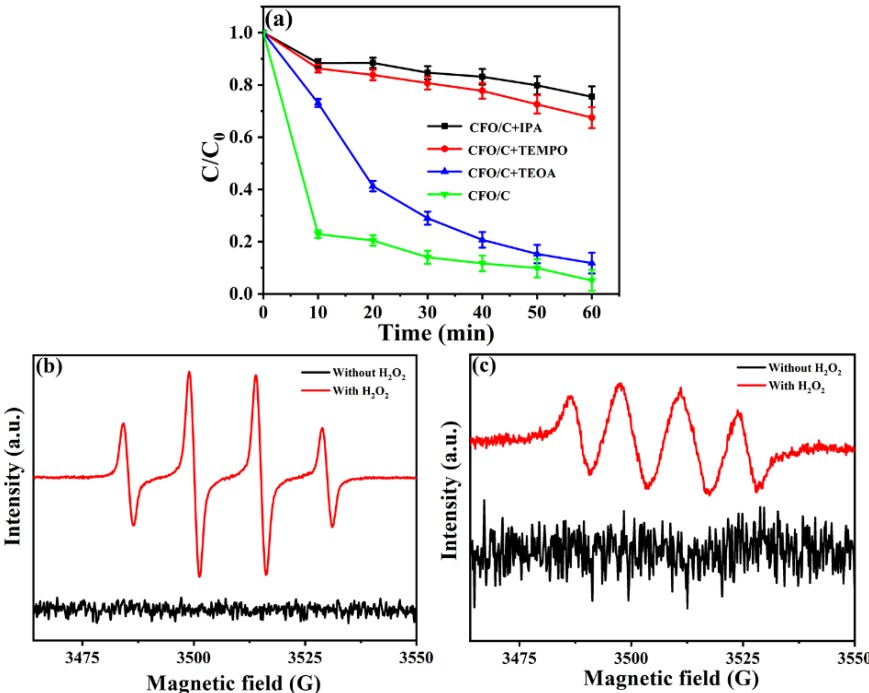

**Figure 11.** (**a**) The effects of various scavengers on photocatalytic activity of CFO/C-PNSs in photo-Fenton degradation condition; (**b**,**c**) the electron spin response (ESR) spectra.

Based on the above characterization analysis and discussion of the experimental results, a possible charge separation mechanism for degradation with the CFO/C-PNS sample is postulated in Figure 12. In this experiment, $\bullet O_2^-$ and $\bullet OH$ radicals are the primary reactive species, resulting from the hole and free-radical trapping tests. Under visible light, excited CFO/C-PNs produce electrons in the conduction band (CB) and holes in the valence band (VB). Subsequently, photoinduced holes could quickly oxidize $H_2O_2$ to $\bullet O_2^-$ because the CB potential of $CuFe_2O_4/C$ ($-0.71$ V) is more positive than the redox potential of $H_2O_2/\bullet O_2^-$ (0.93 V vs. NHE). Meanwhile, photogenerated electrons quickly react with $Fe^{3+}$ to $Fe^{2+}$ and the formed $Fe^{2+}$ can immediately and directly react with $H_2O_2$ to produce hydroxyl radicals ($\bullet OH$). Therefore, $\bullet OH$ was detected in the photo-Fenton system. On the other hand, the holes could directly react with water ($H_2O$) or hydroxyl ions ($OH^-$) to $\bullet OH$; furthermore, electrons could react with dissolved $O_2$ to generate superoxide radicals ($\bullet O_2^-$). More importantly, the recombination of electrons and holes could be effectively separated due to the pomegranate-like CFO/C-PNS structure assembled by ultra-small $CuFe_2O_4/C$ subunits, and each ultrafine $CuFe_2O_4/C$ subunit could effectively and directly participate in the reaction, where a large number of $Fe^{3+}/Fe^{2+}$ and $H_2O_2$ exist to produce $\bullet OH$, and generated contaminant could also be rapidly decomposed to the intermediates. Finally, all these intermediates could be attacked by the $\bullet OH/\bullet O_2^-$ species until the final mineralization.

$$CFO/C\text{-}PNSs + h\nu \rightarrow CuFe_2O_4 \ (e^- + h^+) \tag{1}$$

$$Fe^{3+} + e^- \rightarrow Fe^{2+} \tag{2}$$

$$H_2O_2 + Fe^{2+} \rightarrow Fe^{3+} + \bullet OH + OH^- \tag{3}$$

$$e^- + O_2 \rightarrow \bullet O_2^- \tag{4}$$

$$e^- + H_2O_2 \rightarrow \bullet OH + OH^- \tag{5}$$

$$h_{VB}^+ + H_2O/OH^- \rightarrow H^+ + \bullet OH \tag{6}$$

$$\bullet OH/\bullet O_2^- + \text{degradation products} \rightarrow CO_2 + H_2O \tag{7}$$

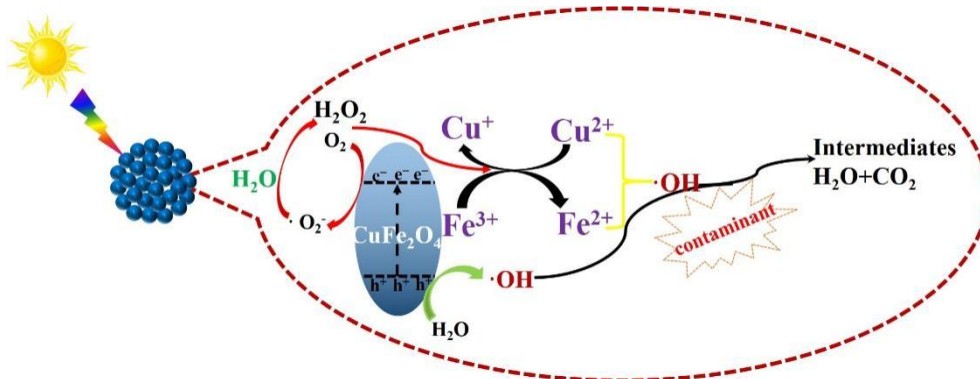

**Figure 12.** Schematic diagram of possible mechanism of photo-Fenton-like catalytic reaction of CFO/C-PNSs.

## 4. Conclusions

In summary, a novel pomegranate-like CFO/C-PNS photo-Fenton catalyst was successfully fabricated by applying a straightforward and cost-effective method. The CFO/C-PNSs exhibited enhanced and persistent photo-Fenton-like reaction behaviors for the degradation and mineralization of oxytetracycline (OTC), norfloxacin (NFX), tetracycline (TCH), rhodamine B (RhB), methyl orange (MO) and Cr(VI) (Cr(VI)) in solution under visible light. The excellent photocatalytic performance could be assigned to its unique nanostructure, which indicates that the integration of the pomegranate-like CFO/C-PNSs composed of ultrafine subunits is a viable way to enhance the photo-Fenton degradation performance.

This work offers a novel strategy to synthesize photocatalytic materials with a pomegranate-like structure and develop the next-generation of a high-performance photo-Fenton catalyst with a wide prospect to facilitate the practical utilization of photodegradation.

**Supplementary Materials:** The following supporting information can be downloaded at: https://www.mdpi.com/article/10.3390/catal12070687/s1, Figure S1: The SEM (a), TEM (b) and HRTEM (c) images for CFO-PNSs sample; Figure S2: The removal rate of (a) CFO/C-PNSs and (b) CFO-PNSs for Cr6+ MO, OTC, NFX, TCH and RhB; Figure S3: The photocatalytic degradation of (a) Cr6+, (b) MO, (c) OTC, (d) NFX, (e) TCH and (f) RhB by CFO/C-PNSs with different conditions under visible light irradiation; Table S1: The comparison of TCH photo-Fenton degradation activity of CFO/C-PNSs with previous literatures [84–103].

**Author Contributions:** Conceptualization, Z.C. and K.L.; methodology, D.S.; formal analysis, D.S., J.Y., F.C., Z.C. and K.L.; investigation, D.S. and J.Y.; resources, Z.C.; data curation, J.Y. and F.C.; writing—original draft preparation, D.S.; writing—review and editing, Z.C. and K.L.; supervision, Z.C. and K.L.; project administration, Z.C. and K.L.; funding acquisition, Z.C. and K.L. All authors have read and agreed to the published version of the manuscript.

**Funding:** The present work was financially supported by the National Natural Science Foundation of China (21801091), the Science and Technology Department project of Jilin Province (YDZJ202201ZYTS591, 20210509049RQ and 20200403019SF) and the Science and Technology Program of Jilin City (20190104169). The authors acknowledge the assistance of the JLICT Center of Characterization and Analysis.

**Data Availability Statement:** Not applicable.

**Conflicts of Interest:** There are no conflicts of interest to declare.

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
