# Peer review of "Hollow Nanospheres Organized by Ultra-Small CuFe2O4/C Subunits with Efficient Photo-Fenton-like Performance for Antibiotic Degradation and Cr(VI) Reduction"

_catalysts, doi:10.3390/catal12070687_

Round 1

Reviewer 1 Report

This work is conducted to fabricate uniform highly dispersible hollow CFO/C-PNSs assembled by ultra-small CFO/C subunits through a simple and cost-effective strategy. The as-synthesized CFO/C-PNSs exhibited markedly photocatalytic performance in photo-Fenton-like reactions for the heterogeneous activation of H2O2 with irradiation of visible light, which can degrade and mineralize oxytetracycline (OTC), norfloxacin (NFX), tetracycline (TCH), rhodamine B (RhB), methyl orange (MO) and Cr(VI) (Cr(VI)) reduction in solution.

The work is well organized with clear results and justified conclusion, herewith it could ne accepted in its present form.

Author Response

Thank you for your approval about this manuscript.

Reviewer 2 Report

Water remediation with degradation of dissolved substances, like drugs, for example, requires efficient treatments. Oxidation processes are preferred in it. The Fenton reaction is one of them. One of the drawbacks of classical Fenton reaction is the use of a large amount of H2O2. It could be reduced with modification of the process going towards the photo-Fenton reaction. A recently published paper, Catalyst, 2022, 12, 257, confirms this direction. Authors should refer to the above manuscript as a background for their interesting paper, which is somehow a natural extension of the mentioned paper.  Authors have designed the composite consisting of primary CuFe2O4/C forms, marked in the text as CFO/C-PNss, for photo-degradation of selected antibiotics, dyes, and reduction of Cr ions, at the photo-Fenton reaction.

The Paper is well organized, and each step of the procedure is clearly explained. Catalyst nanostructure obtained in the process is effective for the assumed condition. The main important question is how the obtained results and defined procedure could be used for optimization and scaling-up of the process.

An important feature of any research paper is the presentation of the possibility of extending the application of its results. The key parameters in the discussed process are mass transport and the harvesting of light quanta in the area of catalytic reaction. These two factors are a derivative of the nanocomposite structure, i.e. the pore size distribution and the distribution of active catalytic forms of CuFe2O4 in the CFO/C-PNss structure.

Authors should add in the text how such a structure can be controlled and what process parameters determine it. This information will raise the scientific level of the article.

Author Response

Reviewer #2:

  1. Water remediation with degradation of dissolved substances, like drugs, for example, requires efficient treatments. Oxidation processes are preferred in it. The Fenton reaction is one of them. One of the drawbacks of classical Fenton reaction is the use of a large amount of H2O2. It could be reduced with modification of the process going towards the photo-Fenton reaction. A recently published paper, Catalyst, 2022, 12, 257, confirms this direction. Authors should refer to the above manuscript as a background for their interesting paper, which is somehow a natural extension of the mentioned paper. Authors have designed the composite consisting of primary CuFe2O4/C forms, marked in the text as CFO/C-PNss, for photo-degradation of selected antibiotics, dyes, and reduction of Cr ions, at the photo-Fenton reaction.

Answer: Thank you for the kind suggestion. After perusing the paper of Catalyst, 2022, 12, 257, we added the relevant description in the view of “One of the drawbacks of classical Fenton reaction is the use of a large amount of H2O2, it could be reduced with modification of the process going towards the photo-Fenton reaction” in the revised manuscript as follows:

“Nevertheless, the traditional homogenous Fenton processes include obvious and inevitable shortcomings, such as production of iron containing sludge, low pH (pH Ë‚ 3.0), a large amount of H2O2, requirement for secondary treatment and difficult regeneration of the catalyst, which have greatly restricted their wide applications [26]. To solve this problem, “Fenton-like” reaction, which can surmount the above-mentioned adverse conditions, such as H2O2 could be reduced with modification of the process going towards the photo-Fenton reaction, has been considered as one of the most effective methods [26-34].”

  1. Simona, F.; Corrado, B.; Sebania, L.; Leon, G.; Daniela, I.; Silvia, S.; Photo-Fenton degradation of methyl orange with dunino halloysite as a source of iron heterogeneous. Catalysts. 2022, 527.
  2. Qian, X.; Ren, M.; Zhu, Y.; Yue, D.; Han, Y.; Jia, J.; Zhao, Y. Visible Light Assisted Heterogeneous Fenton-like Degradation of Organic Pollutant via α-FeOOH/Mesoporous Carbon Composites. Environ. Sci. Technol 2017, 51, 3993-4000.
  3. Jin, H.; Tian, X.; Nie, Y.; Zhou, Z.; Yang, C.; Li, Y.; Lu, L. Oxygen vacancy promoted heterogeneous fentonlike degradation of ofloxacin at pH 3.2-9.0 by Cu substituted magnetic Fe3O4@FeOOH nanocomposite. Environ. Sci. Technol. 2017, 51, 12699-12706.
  4. Khandarkhaeva, M.; Batoeva, A.; Aseev, D.; Sizykh, M.; Tsydenova, O. Oxidation of atrazine in aqueous media by solar-enhanced Fenton-like process involving persulfate and ferrous ion. Ecotoxicol. Environ. Saf. 2017, 137, 35-41.
  5. Fei, B.; Deng, N.; Wang, J.; Liu, Q.; Long, J.; Li, Y.; Mei, X. A heteropoly blue as environmental friendly material: an excellent heterogeneous Fenton-like catalyst and flocculent. J. Hazard. Mater. 2017, 340, 326-335.
  6. Li, X.; Zhang, Y.; Xie, Y.; Zeng, Y.; Li, P.; Xie, T.; Wang, Y. Ultrasonic-enhanced Fenton-like degradation of bisphenol A using a bio-synthesized schwertmannite catalyst. J. Hazard. Mater. 2018, 344, 689-697.
  7. Koltsakidou, Α.; Antonopoulou, M.; Sykiotou, M.; Εvgenidou, Ε.; Konstantinou, I.; Lambropoulou, D.A. Photo-Fenton and Fenton-like processes for the treatmentof the antineoplastic drug 5-fluorouracil under simulated solar radiation. Environ. Sci. Pollut. Res. 2017, 24, 1-10.
  8. Mao, J.; Quan, X.; Wang, J.; Gao, C.; Chen,S.; Yu, H.; Zhang, Y. Enhanced heterogeneous Fenton-like activity by Cu-doped BiFeO3 perovskite for degradation of organic pollutants. Front. Environ. Sci. Eng. 2018, 12, 103-112
  9. Zhou, L.; Xu, Z.; Zhang, J.; Zhang, Z.; Tang, Y. Degradation of hydroxypropyl guar gum at wide pH range by a heterogeneous Fenton-like process using bentonite-supported Cu(0). Prog. Water Technol. 2020, 82, 1635-1642.

  1. The Paper is well organized, and each step of the procedure is clearly explained. Catalyst nanostructure obtained in the process is effective for the assumed condition. The main important question is how the obtained results and defined procedure could be used for optimization and scaling-up of the process.

Answer: Thank you for the kind suggestion. Currently, the laboratory is enclosed due to the severe epidemic situation. Therefore, we failed to carry out the experiments of optimization and scaling-up of the process. However, we will focus on this issue in the future.

  1. An important feature of any research paper is the presentation of the possibility of extending the application of its results. The key parameters in the discussed process are mass transport and the harvesting of light quanta in the area of catalytic reaction. These two factors are a derivative of the nanocomposite structure, i.e. the pore size distribution and the distribution of active catalytic forms of CuFe2O4 in the CFO/C-PNss structure.

Authors should add in the text how such a structure can be controlled and what process parameters determine it. This information will raise the scientific level of the article.

Answer: Thank you for the kind suggestion. The preparation process of composite materials is presented as follows:

“Firstly, PAA-NH4 with a globose structure was prepared by addition of NH3•H2O and isopropyl alcohol aqueous solution. After that, a certain amount of FeCl2•4H2O was added into the PAA-NH4 aqueous solution and Fe2+ was further anchored to the surface of an anionic polymer (PAA) at random to form Fe(OH)3/PAA solution in the alkaline conditions due to the charge of the interfacial energy of the synthetic system. Then, CuCl2•2H2O was added, Cu2+ readily hydrolyzed into Cu(OH)2 nanoparticles in a weak base environment, and further aggregated on the outer and inner surfaces of Fe(OH)3/PAA layer to synthesize Fe(OH)3/PAA/Cu(OH)2 nanoparticles. More importantly, due to the electrostatic repulsion of the carboxyl groups, the Fe(OH)3/PAA/Cu(OH)2 in solution has high dispersibility. Lastly, the CuFe2O4/C-PNSs with hollow nanostructures were fabricated by heating treatment at 400 ℃ for 120 min under N2 gas protection, and named as CFO/C-PNSs.”

It can be seen that the synthesis process is non-toxic, simple and easy to operate. We think the key process parameters are pH, calcination temperature and calcination time.

Furthermore, the unique hollow spherical CuFe2O4/C-PNSs assembled by CuFe2O4/C subunit have a high specific surface area 179.7 m2/g and excellent mass transfer performance, which is beneficial to improve the photocatalytic performance. The high specific surface area provides more active sites and allows more pollutant molecules to adsorb. Meanwhile, hollow microspheres enhance the utilization efficiency of incident visible light by enhancing the reflection and scattering of light. Therefore, CuFe2O4/C-PNSs catalyst present outstanding performance towards degrading a variety of antibiotics and dyes.

Reviewer 3 Report

In this paper, the authors report on uniformly and highly dispersed hollow nanospheres prepared by a one-pot approach used for antibiotic and dye degradation under visible light irradiation.

Even though the overall research design and results are extensive and mostly well-presented, at least the following should be addressed:

1. Throughout the entire manuscript writing has to be improved, especially the syntax. That would greatly improve the readability of the manuscript because, at this form, it is really hard to read.

2. The authors should write the complete terms of all abbreviations in the abstract.

3. In the introduction part, I would suggest that authors put an emphasis on the innovation and application of their material, especially due to synthesis novelty. Also, a better comparison with similar work should be elaborated.

4. In the Experimental part, the sentence “Lastly, the CFO/CPNSs with hollow nanostructures were fabricated by heating treatment at 400 ℃ for 120 min under N2 gas protection, and named as CFO-PNSs“ is problematic because written this way, it is not clear are the CFO/CPNS or CFO-PNS synthesized after heating treatment. That being said, the syntax is a major drawback of this manuscript.

4. Results and discussion part is thorough, photocatalysts are well-characterized. I would suggest that the authors write what was the temperature of the photocatalytic experiments. 

Author Response

Reviewer #3: In this paper, the authors report on uniformly and highly dispersed hollow nanospheres prepared by a one-pot approach used for antibiotic and dye degradation under visible light irradiation.

Even though the overall research design and results are extensive and mostly well-presented, at least the following should be addressed:

  1. Throughout the entire manuscript writing has to be improved, especially the syntax. That would greatly improve the readability of the manuscript because, at this form, it is really hard to read.

Answer: Thank you for the kind suggestion. We have tried our best to improve the English level of the manuscript.

  1. The authors should write the complete terms of all abbreviations in the abstract.

Answer: Thank you for the kind suggestion. In the revised manuscript, we added the complete terms of all the abbreviations in the abstract as follows:

“Scanning electron microscope (SEM) and transmission electron microscope (TEM) images verified that CFO/C-PNSs catalyst mainly presents hollow nanospheres morphology with diameter of 250 ± 30 nm.”

  1. In the introduction part, I would suggest that authors put an emphasis on the innovation and application of their material, especially due to synthesis novelty. Also, a better comparison with similar work should be elaborated.

Answer: Thank you for the kind suggestion. We have highlighted the innovation and application of the material, especially the synthesis novelty in the revised manuscript as follows:

“In this work, we report one pot synthetic calcination method for the first time to fabricate uniform highly dispersible hollow CFO/C-PNSs assembled by ultra-small CFO/C subunits through a simple and cost-effective strategy. The hollow microspheres have high specific surface area and excellent mass transfer performance, which is beneficial to improve the photocatalytic performance. The high specific surface area provides more active sites and allows more pollutant molecules to adsorb. Meanwhile, hollow microspheres enhance the utilization efficiency of incident visible light by enhancing the reflection and scattering of light. The as-synthesized CFO/C-PNSs exhibited markedly photocatalytic performance in photo-Fenton-like reactions for the heterogeneous activation of H2O2 with irradiation of visible light, which can degrade and mineralize oxytetracycline (OTC), norfloxacin (NFX), tetracycline (TCH), rhodamine B (RhB), methyl orange (MO) and Cr(VI) reduction in solution.”

Secondly, the comparison of photo-Fenton degradation activity for CFO/C-PNSS with previous reports was added as Table S2 in the revised supplementary material as follows:

Table S2. The comparison of photo-Fenton degradation activity of CFO/C-PNSs with previous literatures.

Catalyst

Concentration (mg•L-1)

Dosage

(g•L-1)

Time (min)

pollutant

Removal (%)

Light source

morphology

preparation methods

Reference

CuFe2O4/NC

15

0.3

90

Levofloxacin (LVFX)

84.87%

nanoparticle

solvothermal method

17

CuFe2O4

20

0.16

150

Malachite Green (MG)

82.8%

300 W Xenon lamp

Porous and spongy

solution combustion synthesis

18

oxygen vacancies-

CuFe2O4

10

0.1

90

sulfamethazine (SMT)

95%

500 W Xenon lamp; λ > 420 nm

raw expanded perlite

precipitation-calcination method

19

CuFe2O4/AgBr

16.34

1.51

120

acid red 88 (AR88)

94.7%

300 W Xenon lamp

particle

hydrothermal method

20

CuFe2O4/g-C3N4

103.7

1

120

propranolol (PRO)

82.2%

350W Xenon lamp; λ > 420 nm

layered structure

sol-gel combustion

method

21

Ag3PO4/

NrGO/

CuFe2O4

15

0.3

60

2,4-

dichlorophenol (2,4-DCP)

95.3%

Xe lamp 250 W; λ ≤ 420 nm

tetrahedronnanoparticls morphology

Coprecipitation method

22

Bi2Te3/

CdS/

CuFe2O4

40

1.4

180

Methylene Blue (MB)

97.15%

ungsten-

halogen lamp

Nanorod with anoparticles

wet impregnation method

23

CuFe2O4/ C

20

0.6

60

rhodamine B (RhB)

99.1%

350W Xenon lamp

Hollow nanospheres

one pot synthetic calcination method

This work

  1. Dong, Z.; Niu, C.; Guo, H.; Niu, H.; Liang, S.; Liang, C.; Liu, H.; Yang, Y. Anchoring CuFe2O4 nanoparticles into N-doped carbon nanosheets for peroxymonosulfate activation: Built-in electric field dominated radical and non-radical process. Chem. Eng. J. 2021, 426, 130850.
  2. Shetty, K.; Renuka, L.; Nagaswarupa, H.; Nagabhushana, H.; Anantharaju, K.; Rangappa, D.; Prashantha, S.; Ashwini, K. A comparative study on CuFe2O4, ZnFe2O4 and NiFe2O4: Morphology, Impedance and Photocatalytic studies. Mater. Today: Proc. 2017, 4, 11806-11815.
  3. Sun, Q.; Wang, X.; Liu, Y.; Xia, S.; Zhao, J. Activation of peroxymonosulfate by a floating oxygen vacancies-CuFe2O4 photocatalyst under visible light for efficient degradation of sulfamethazine. Sci. Total Environ. 2022, 824, 153630.
  4. Zhang, X.; Zhao Y. Optimization of photocatalytic degradation of dye wastewater by CuFe2O4/AgBr composite using response surface methodology. Mater. Res. Express. 2018, 6, 036109.
  5. Li, R.; Cai, M.; Xie, Z.; Zhang, Q.; Zeng, Y.; Liu, H.; Liu, G.; Lv, W. Construction of heterostructured CuFe2O4/g-C3N4 nanocomposite as an efficient visible light photocatalyst with peroxydisulfate for the organic oxidation. Appl. Catal., B. 2019, 244, 974-982.
  6. Wei, X.; Yang, X.; Xu, X.; Liu, Z.; Naraginti, S.; Wan, J. Novel magnetically separable tetrahedral Ag3PO4/NrGO/CuFe2O4 photocatalyst for efficient detoxification of 2,4-dichlorophenol. Environ. Res. 2021, 201, 111519.
  7. Palanisamy, G.; Bhuvaneswari, K.; Bharathi, G.; Pazhanivel, T.; Dhanalakshmi, M. Improved photocatalytic performance of magnetically recoverable Bi2Te3/CdS/CuFe2O4 nanocomposite for MB dye under visible light exposure. Solid State Sci. 2021, 115, 106584.

  1. In the Experimental part, the sentence “Lastly, the CFO/C-PNSs with hollow nanostructures were fabricated by heating treatment at 400 ℃ for 120 min under N2 gas protection, and named as CFO-PNSs“ is problematic because written this way, it is not clear are the CFO/C-PNSs or CFO-PNSs synthesized after heating treatment. That being said, the syntax is a major drawback of this manuscript.

Answer: Thank you for the kind suggestion. We corrected this issue in the revised manuscript as follows:

“Lastly, the CuFe2O4/C-PNSs with hollow nanostructures were fabricated by heating treatment at 400 ℃ for 120 min under N2 gas protection, and named as CFO/C-PNSs.”

Moreover, we have tried our best to improve the English level in the revised manuscript.

  1. Results and discussion part is thorough, photocatalysts are well-characterized. I would suggest that the authors write what was the temperature of the photocatalytic experiments.

Answer: Thank you for the kind suggestion. The experiment information is described in the supporting information as follows:

“All the photocatalytic degradation experiments were carried out at room temperature, and the photocatalytic activity of different samples was also tested by the photodegradation of antibiotics, using a glass vessel with a water-cooling jacket as reactor (the temperature is about 15 ℃). ”

Reviewer 4 Report

In general, the article describes a rather interesting study on the synthesis of catalysts based on CuFe2O4/C sibunits for the degradation of antibiotics and the reduction of chromium (VI). Questions arise mainly on the design of the article, in particular, the text does not match the template.

Minor comments:

1. Fig. 2. All peaks should be clearly labeled.

2. What explains the almost identical activity in all systems in Figures 5a-e?

3. The main question is about the practical expediency of using hydrogen peroxide for oxidation, this is a rather expensive reagent, and it is known that in the Fenton system only a small part of the peroxide is used specifically for photocatalytic oxidation.

Author Response

Reviewer #4: In general, the article describes a rather interesting study on the synthesis of catalysts based on CuFe2O4/C sibunits for the degradation of antibiotics and the reduction of chromium (VI). Questions arise mainly on the design of the article, in particular, the text does not match the template.

Minor comments:

  1. Fig. 2. All peaks should be clearly labeled.

Answer: Thank you for the kind suggestion. We have re-labeled all peaks of XRD and the results are presented in Fig. 2(a) in the revised manuscript as follows:

Figure 2. XRD spectra of CFO/C-PNSs and CFO-PNSs samples.

  1. What explains the almost identical activity in all systems in Figures 5a-e?

Answer: Thank you for the kind suggestion. As shown in Figure 5 a-e, CFO/C-PNSs and CFO-PNSs did have a difference in degradation performance as follows:

Catalysts Antibiotics

TCH

NFX

OTC

MO

RhB

Cr6+

CFO/C-PNSs

98.75%

97.13%

92.36%

90.60%

99.1%

90.42%

CFO-PNSs

80.35%

87.92%

75.63%

81.16%

84.23%

72.06

The reaction kinetics rates of CFO/C-PNSs and CFO-PNSs towards pollutants degradation also confirmed this issue as follows:

CatalystsAntibiotics

Cr6+

MO

OTC

NFX

TCH

RhB

CFO/C-PNSs

0.02404

0.03138

0.04090

0.04607

0.05183

0.08155

CFO-PNSs

0.02086

0.02677

0.02003

0.03416

0.02366

0.03125

  1. The main question is about the practical expediency of using hydrogen peroxide for oxidation, this is a rather expensive reagent, and it is known that in the Fenton system only a small part of the peroxide is used specifically for photocatalytic oxidation.

Answer: Thank you for the kind suggestion. The comparison of hydrogen peroxide dosage for the CFO/C Fenton system with previous reports is presented in Table R1, and we think the dosage of H2O2 in our work is moderate.

Table R1. The comparison of hydrogen peroxide dosage for the CFO/C Fenton system with previous reports.

Catalyst

Concentration (mg·L-1)

Dosage (g·L-1)

Time (min)

Pollutant

Removal (%)

H2O2 Dosage

(mM)

Reference

Cu(II)

9.9

0.0032

120

2,4,6-

trichlorophenol (TCP)

70

60

1

Fe3O4

4.8

0.5

120

RhB

98

40

2

Fe2+

1000

0.007

120

Phenol

96%

14.7

3

Cu(0)

5

6

45

Hydroxypropyl
guar gum (HPGG)

76

17.7

4

CFO/C

20

0.6

60

rhodamine B (RhB)

99.1

21.6

This work

  1. Wang, Z.; Liu, Q.; Yang, Fei.; Huang, Y.; Xue,Y.; Yuan, R.; Sheng, Bo.; Wang, X. Accelerated oxidation of 2,4,6-trichlorophenol in Cu(II)/H2O2/Cl- system: A unique “halotolerant” Fenton-like process. Environment international, 2019,132, 105128.
  2. Chen, F.; Xie,S.; Huang, X.; Qiu, X.; Ionothermal synthesis of Fe3O4 magnetic nanoparticles as efficient heterogeneous Fenton-like catalysts for degradation of organic pollutants with H2O2. J. Hazard. Mater. 2017, 322,152-162.
  3. Messele, S.; Bengoa, C.; Stüber, F.; Giralt, Jaume.; Fortuny, A.; Fabregat, A.; Font, J.; Enhanced Degradation of Phenol by a Fenton-Like System (Fe/EDTA/H2O2) at Circumneutral pH. Catalysts. 2019, 9, 474.

Zhou, L.; Xu, Z.; Zhang, J.; Zhang, Z.; Tang, Y. Degradation of hydroxypropyl guar gum at wide pH rangeby a heterogeneous Fenton-like process using bentonite-supported Cu(0). Water Sci. Technol. 2020, 82, 1635-16